# Pharmacokinetic/Pharmacodynamic Analysis of Continuous-Infusion Fosfomycin in Combination with Extended-Infusion Cefiderocol or Continuous-Infusion Ceftazidime-Avibactam in a Case Series of Difficult-to-Treat Resistant *Pseudomonas aeruginosa* Bloodstream Infections and/or Hospital-Acquired Pneumonia

**DOI:** 10.3390/antibiotics11121739

**Published:** 2022-12-02

**Authors:** Milo Gatti, Maddalena Giannella, Matteo Rinaldi, Paolo Gaibani, Pierluigi Viale, Federico Pea

**Affiliations:** 1Department of Medical and Surgical Sciences, Alma Mater Studiorum University of Bologna, 40138 Bologna, Italy; 2Clinical Pharmacology Unit, Department for Integrated Infectious Risk Management, IRCCS Azienda Ospedaliero-Universitaria di Bologna, 40138 Bologna, Italy; 3Infectious Diseases Unit, Department for Integrated Infectious Risk Management, IRCCS Azienda Ospedaliero-Universitaria di Bologna, 40138 Bologna, Italy; 4Microbiology Unit, IRCCS Azienda Ospedaliero-Universitaria di Bologna, 40138 Bologna, Italy

**Keywords:** fosfomycin, DTR-*Pseudomonas aeruginosa*, continuous infusion, cefiderocol, ceftazidime-avibactam, TDM-guided strategy, PK/PD, combination therapy, microbiological eradication

## Abstract

Objectives: To perform a pharmacokinetic/pharmacodynamic (PK/PD) analysis of continuous-infusion (CI) fosfomycin combined with extended-infusion (EI) cefiderocol or CI ceftazidime-avibactam in a case series of severe difficult-to-treat Pseudomonas aeruginosa (DTR-PA) infections. Methods: A single-center retrospective study of patients who were treated with CI fosfomycin plus EI cefiderocol or CI ceftazidime-avibactam for severe DTR-PA infections and who underwent therapeutic drug monitoring (TDM), from 1 September 2021 to 30 June 2022 was performed. Concentrations were measured at steady-state (C_ss_) for CI fosfomycin and ceftazidime-avibactam and at trough (C_min_) for EI cefiderocol. Joint PK/PD targets of combination therapy were analyzed (thresholds: area-under-the curve to minimum inhibitory concentration (AUC/MIC) ratio > 40.8 for fosfomycin; ceftazidime C_ss_/MIC ratio ≥ 4 coupled with avibactam C_ss_ > 4 mg/L for ceftazidime-avibactam; C_min_/MIC ratio ≥ 4 for cefiderocol). Joint PK/PD targets of the combination therapy were analyzed and defined as optimal when both were achieved, quasi-optimal if only one of the two was achieved, and suboptimal if none of the two was achieved). The relationship between joint PK/PD target attainment and microbiological response was assessed. Results: Six patients (three pneumonia, two BSI + pneumonia, and one BSI) were included. The joint PK/PD targets were optimal in four cases and quasi-optimal in the other two. Microbiological eradication (ME) occurred in 4/4 of patients with optimal joint PK/PD targets and in one of the two patients with quasi-optimal joint PK/PD targets. Conclusions: Attaining optimal joint PK/PD targets with a combo-therapy of CI fosfomycin plus EI cefiderocol or CI ceftazidime-avibactam could represent an effective strategy for granting favorable microbiological outcomes in patients with DTR-PA pneumonia and/or BSI.

## 1. Introduction

The widespread diffusion of multidrug-resistant (MDR) Gram-negative pathogens is a worrisome health concern, representing one of the main causes of hospital morbidity and mortality [1]. *Pseudomonas aeruginosa* (PA) is responsible for a remarkable number of severe infections (particularly pneumonia) in hospitalized critically ill patients [2]. Other than being intrinsically resistant to several antimicrobials, PA may easily develop new resistance mechanisms, thus, becoming MDR [3]. Recently, the definition of “difficult-to-treat resistance” (DTR) was proposed for identifying those strains of PA that are simultaneously resistant to carbapenems, third/fourth-generation cephalosporins, piperacillin-tazobactam, aztreonam, and fluoroquinolones [4]. Consequently, choosing an effective antibiotic therapy for managing DTR-PA infections has become extremely challenging. Several international guidelines have recently proposed the use of novel beta-lactams (BLs) and/or beta-lactams/beta-lactamase inhibitor combinations (BL/BLIc) (i.e., ceftolozane-tazobactam, ceftazidime-avibactam, imipenem-relebactam, and cefiderocol) as first-line therapy for the management of severe DTR-PA infections [5,6]. However, monotherapy with these agents is not always appropriate in all of the clinical scenarios.

Fosfomycin is an old antibiotic that was recently repurposed as a potential add-on treatment for MDR Gram-negative infections. Fosfomycin has a broad spectrum of activity, excellent penetration into deep-seated infections, and a good safety profile [7]. Preclinical models of DTR-PA infections showed that the combination of fosfomycin with beta-lactams was superior to either drug alone, with synergism reported with carbapenems in 73.3–100% of isolates [8,9,10]. Additionally, in a murine infection model using a high bacterial burden, combination therapy including ceftazidime-avibactam and fosfomycin significantly reduced the DTR-PA colony-forming units by approximately 2 and 5 logs compared with stasis and in the vehicle-treated control, respectively [11]. Consequently, the combinations of fosfomycin with ceftazidime-avibactam but also with cefiderocol could be a promising option for managing severe DTR-PA infections. Other than for this purpose, it should not be overlooked that during the pandemic era, a long-term supply shortage of ceftolozane-tazobactam occurred [12]. Consequently, in the last two years or so, this approach was increasingly used also as an alternative therapeutic strategy to ceftolozane-tazobactam for treating DTR-PA infections.

Optimizing antimicrobial exposure according to accepted pharmacokinetic/pharmacodynamic (PK/PD) properties is considered fundamental by several guidelines nowadays [13,14]. In regard to BLs and/or BL/BLIc, administration by extended (EI) or continuous infusion (CI) may provide significant advantages over intermittent infusion in terms of achieving more aggressive pharmacokinetic/pharmacodynamics (PK/PD) targets and favorable clinical outcome [15,16]. In regard to fosfomycin, emerging evidence suggested that CI could be the best therapeutic strategy for maximizing PK/PD targets for preventing resistance emergence among patients with MDR Gram-negative infections [17,18,19].

A therapeutic drug monitoring (TDM)-guided personalized antimicrobial dosing with expert clinical pharmacological interpretation could represent a useful tool for promptly maximizing the attainment of optimal PK/PD targets in this challenging scenario. To this regard, different real-world studies recently highlighted the clinical relevance of a TDM-guided approach for personalized tailored therapy with novel BLs and/or BL/BLIc in deep-seated infections [20,21,22,23,24,25]. However, real-world data testing of the effectiveness of fosfomycin in combination with novel BLs or BL/BLIc in DTR-PA infections in relation to PK/PD target attainment is currently lacking [5,6].

The aim of this study was to carry out a PK/PD analysis of combination therapy with CI fosfomycin plus EI cefiderocol or CI ceftazidime-avibactam in a case series of DTR-PA bloodstream infections (BSIs) and/or hospital-acquired pneumonia (HAP).

## 2. Results

Overall, during the study period, six patients with documented severe DTR-PA infections were treated with CI fosfomycin in combination with EI cefiderocol (*n* = 4) or CI ceftazidime-avibactam (*n* = 2). Demographic and clinical features are summarized in Table 1.

The mean (±standard deviation [SD]) age was 57.7 ± 21.7 years with a male preponderance (67%). Five out of six patients (83%) were ICU admitted. Types of infection were VAP in two cases, HAP in one case, bloodstream infection (BSI) in one case, and VAP plus BSI in the other two cases. The susceptibility profile of the DTR-PA clinical isolates and the rationales adopted for selecting the combination therapy of fosfomycin with ceftazidime-avibactam or cefiderocol are summarized in Table 2.

The reasons for not using ceftolozane-tazobactam as the first choice for DTR-PA were supply shortage in four cases and resistance in the other two due to MBL production. In regard to fosfomycin susceptibility of the six DTR-PA isolates, the MIC was 32 mg/L for 3/6, 64 mg/L for 2/6, and 256 mg/L for 1/6.

A maintenance dose (MD) of CI fosfomycin was begun at 16 g/day in 5/6 patients (83.3%) and at 24 g/day in the other one (17%). In regard to CI of ceftazidime-avibactam, MD was started with 2.5 g q8h over 8 h in both cases. EI of cefiderocol was started at 2 g q8h over 3 h in three patients and at the intensified dose of 2 g q6h over 3 h in another one. The median (interquartile range [IQR]) average *f*C_ss_ of fosfomycin, ceftazidime, and avibactam were 504.9 mg/L (363.2–647.2 mg/L), 83.1 mg/L (65.2–100.9 mg/L), and 18.0 mg/L (13.2–22.8 mg/L), respectively. Median (IQR) *f*C_min_ of cefiderocol was 16.3 mg/L (10.3–26.5 mg/L).

The joint PK/PD targets were optimal in four cases (67%) and quasi-optimal in the other two (33%). In these latter cases, PK/PD threshold non-attainment concerned fosfomycin in one case and cefiderocol in the other. Microbiological eradication was obtained in all but one case (5/6; 83%). Microbiological failure occurred in a VAP patient (17%) with a quasi-optimal joint PK/PD target (due to fosfomycin). Real-time TDM guided dosing adaptation was performed in two cases for fosfomycin (33%; all reductions), in two cases for cefiderocol (50%; one reduction and one increase), and in one case for ceftazidime/avibactam (50%; one reduction, Table 1).

No antimicrobial treatment-related adverse event emerged. The overall 30-day mortality rate was 33%, but both of the patients were deceased for underlying diseases unrelated to DTR-PA infections.

## 3. Discussion

To the best of our knowledge, this is the first case series that has analyzed the relationship between the PK/PD profile of the combination of CI fosfomycin with EI cefiderocol or CI ceftazidime-avibactam and the microbiological outcome of DTR-PA related BSI and/or HAP/VAP. Our case series, although limited in size, suggests that attaining aggressive joint PK/PD targets with these combination therapies by means of a real-time TDM-guided approach could be an effective strategy when facing against DTR-PA BSI and or HAP/VAP.

The theoretical role of combination therapy in treating DTR-PA infections is still a matter of debate. Neither the ESCMID [6] nor the IDSA guidelines [5] provide any recommendation toward combination therapy either in favor or against. Interestingly, some preclinical models showed that fosfomycin may have a synergic or additive effect with ceftazidime-avibactam against DTR-PA [11,26,27], whereas no study has yet assessed the effect of combining fosfomycin with cefiderocol. Overall, clinical data on the efficacy of both of these combinations in treating DTR-PA infections are lacking.

Our study first showed that both of these combinations may be effective in DTR-PA infections when attaining an optimal joint PK/PD target. Microbiological eradication occurred in all of our BSI cases and in 80% of HAP/VAP cases. The choice of using ceftazidime/avibactam or cefiderocol for treating DTR-PA was due to the impossibility of using ceftolozane-tazobactam because of supply shortage and/or in vitro resistance. Administration by EI for cefiderocol and by CI for ceftazidime/avibactam were pursued for attaining very aggressive PK/PD targets, which were shown to grant significant advantages in terms of microbiological and/or clinical outcome in severe Gram-negative infections [15,16,28,29,30]. Subsequently, implementing a TDM-guided strategy allowed us to tailor the antibiotic dosage in each single patient for maximizing PK/PD target attainment while avoiding unnecessary overexposure. Indeed, although no treatment-related adverse event emerged, we adopted this conservative approach because we are well aware that some traditional BLs may have exposure thresholds for neurotoxicity risk [31], and the same could not be ruled out yet for novel BLs. The choice of adding fosfomycin as combination therapy was based on the rationale of seeking potential synergic or additive effects with BL against DTR-PA [11,32]. Additionally, for pneumonia cases, it was thought that the high penetration rate of fosfomycin in the epithelial lining fluid [ELF] [33] would be beneficial for attaining more aggressive PK/PD targets, even at the infection site. Administration of fosfomycin by CI could be the best strategy for both maximizing PK/PD target attainment and preventing resistance emergence among MDR Gram-negative infections [17,18,19]. In this regard, we showed in a previous case of carbapenem-resistant PA ventriculitis treated with CI fosfomycin in combination with CI ceftazidime/avibactam that the penetration rate of fosfomycin into the CSF was as high as 50% and that attaining optimal joint PK/PD targets at the infection site was fundamental for granting microbiological eradication and clinical success [20,21,22,23,24,25].

Notably, EUCAST stated that there are not enough data for providing any meaningful clinical breakpoint of fosfomycin against PA. Accordingly, defining which PK/PD target would be achievable against DTR-PA with standard dosages of CI fosfomycin (16–24 g/die) would be very informative for clinicians. In this regard, the findings suggest that our TDM-guided approach may allow us to deal effectively with DTR-PA strains with an MIC up to 64 mg/L. Notably, it should be mentioned that the only case with microbiological failure in our series was that of a VAP patient treated with fosfomycin in combination with ceftazidime-avibactam in whom the optimal PK/PD target of fosfomycin was not attained because of a very high fosfomycin MIC value (256 mg/L). Additionally, the attained PK/PD target of ceftazidime/avibactam was optimal but borderline and this might have concurred in failure. The penetration rate of ceftazidime/avibactam into the ELF is approximatively 30% [34], and previous real-world studies found that pneumonia may be an independent risk factor for clinical failure and mortality among patients treated with ceftazidime-avibactam [28,35].

We recognize that our study has some limitations. The retrospective monocentric design and the limited sample size should be acknowledged. As total drug concentrations were measured, the free moieties were only estimated. Conversely, the analysis of the relationship between the joint PK/PD target attainment of combination therapy and the microbiological outcome of severe DTR-PA infections is a point of strength.

In conclusion, our findings suggest that administering fosfomycin by CI as part of a combination therapy including CI ceftazidime-avibactam or EI cefiderocol and adopting a strategy of real-time TDM-guided dosing adaptation may be very helpful in attaining optimal joint PK/PD targets in challenging scenarios of DTR-PA infections. This approach may lead to microbiological eradication in most cases of DTR-PA BSI and/or pneumonia with limited therapeutic options. Large prospective studies are warranted for confirming our hypothesis.

## 4. Materials and Methods

This retrospective study included a case series of patients who were treated with CI fosfomycin in combination with CI ceftazidime-avibactam or EI cefiderocol for documented severe DTR-PA infections and underwent real-time TDM of these antimicrobials at the IRCCS Azienda Ospedaliero-Universitaria of Bologna between 1 September 2021 and 30 June 2022.

Demographic and clinical/laboratory data were retrieved for each single patient. The type/site of infection, fosfomycin and beta-lactam dosages, treatment duration, MIC of fosfomycin and of beta-lactams against DTR-PA, and needs for dosing adjustments were also collected. The documented BSI was defined as the isolation of DTR-PA from blood cultures. The documented HAP was defined as the isolation of a bacterial load of DTR-PA ≥ 10^4^ CFU/mL in the bronchoalveolar lavage (BAL) fluid collected and cultured after >48 h from hospitalization (>48 h from endotracheal intubation and start of mechanical ventilation for VAP) [36,37].

A targeted combination therapy of DTR-PA with cefiderocol or ceftazidime/avibactam plus fosfomycin was prescribed at the discretion of the infectious disease consultant. Treatment was always started with a loading dose (LD) of each drug (8 g over 2 h for fosfomycin; 2.5 g over 2 h for ceftazidime/avibactam, and 2 g over 3 h for cefiderocol). MD regimens were initially chosen on the basis of the patient’s renal function and underlying pathophysiological conditions and, subsequently, optimized by means of adaptive TDM. CI ceftazidime/avibactam was granted by reconstitution of aqueous solutions every 8 h and administration over 8 h.

The antimicrobial susceptibility of the antibacterial agents was tested by gold standard methods as recommended (fosfomycin was tested by means of agar-dilution; ceftazidime-avibactam by means of broth microdilution; cefiderocol by means of broth microdilution coupled with iron-depleted cation-adjusted Mueller–Hinton broth (ID-CAMHB)). MIC values were interpreted according to the EUCAST guidelines [38].

Blood samples for TDM were collected first within 72 h from starting treatment and reassessed whenever feasible. Concentrations of CI fosfomycin, ceftazidime, and avibactam were determined at steady-state (C_ss_), while those of EI cefiderocol were measured at trough (C_min_). The total serum concentrations of each drug were determined by means of validated liquid chromatography-tandem mass spectrometry methods [39,40,41].

As only total concentrations were measured, the free fractions (*f*) of fosfomycin, ceftazidime, avibactam, and cefiderocol were calculated by taking into account the percentage of plasma protein binding reported in the literature (1%, 10%, 7%, and 58%, respectively) [42,43,44]. The AUC/MIC ratio was selected as the PD parameter for best describing fosfomycin efficacy in terms of microbiological outcome. According to preclinical studies of *Pseudomonas aeruginosa* infections, the target *f*AUC/MIC ratio was set at >40.8 [45]. The C_ss_/MIC ratio or the C_min_/MIC ratio was selected as the PD parameter for best describing cefiderocol and ceftazidime efficacy in terms of microbiological outcome and set at ≥4, this value being associated with suppression of resistance emergence [46,47]. In regard to ceftazidime-avibactam, additionally, a C_ss_/threshold concentration (C_T_) ratio for avibactam > 1 (equivalent to 100%*f*T > C_T_ of 4.0 mg/L) was also required for achieving optimal PK/PD targets.

The desired joint PK/PD targets of combination therapy were defined as optimal when both of the agents attained the desired target, as quasi-optimal when only one of the two thresholds was achieved, and suboptimal if none of the two thresholds were achieved. Dosing adjustments were provided on the basis of our current clinical practice, as previously reported [48].

Microbiological failure was defined as the persistence of the same bacterial isolate in blood culture or in BAL culture after ≥7 days from starting antimicrobial combination treatment, as previously reported [35]. Microbiological eradication was defined as the eradication of the original pathogens from the blood or BAL culture of the specimens in at least one subsequent assessment. Follow-up blood cultures (in patients with BSI) and/or BAL cultures (in patients with pneumonia) were executed between day 2 and day 7 and between day 5 and day 14, respectively, to assess microbiological eradication and define treatment duration. The relationship between the joint PK/PD targets and the microbiological outcome was assessed in relation to the site of infection. Secondary outcomes included 30-day mortality rate and occurrence of adverse events (AEs).

Descriptive statistics were used. Continuous data were presented as mean ± SD or median and IQR, whereas categorical variables were expressed as count and percentage. The study was approved by the Ethics Committee of IRCCS Azienda Ospedaliero-Universitaria of Bologna (n. 442/2021/Oss/AOUBo).

## Figures and Tables

**Table 1 antibiotics-11-01739-t001:** Demographic and clinical features of patients with severe infections caused by DTR-PA treated with combination therapy including fosfomycin plus novel beta-lactams.

ID Cases	Age/Sex	Ward	Type of Infection	Fosfomycin MIC (mg/L)	Fosfomycin Dosage	AUC/MIC Ratio (mg/L∙h)	Fosfomycin Dosing Adjustment	Beta-Lactam Co-Treatment	Beta-Lactam MIC (mg/L)	Average *f*C_ss_/MIC Ratio or *f*C_min_/MIC Ratio	Beta-Lactam Dosing Adjustment	Joint PK/PD Target	Microbiological Eradication	30-Day Mortality
DTR *Pseudomonas aeruginosa*														
#1	27/F	Infectious disease unit	HAP	64	8 g LD 16 g/day CI	92.0	No	Cefiderocol 2 g q8h (EI)	1	19.7	No	Optimal	Yes	No
#2	61/F	ICU	VAP	256	8 g LD 16 g/day CI	32.4	No	CAZ-AVI 2.5 g q8h CI	8	5.9 (avibactam *f*C_ss_ 8.4 mg/L)	No	Quasi-optimal	No	Yes
#3	75/M	ICU	BSI + VAP	32	8 g LD 16 g/day CI	471.4	Reduction (12 g/day CI)	Cefiderocol 2 g q8h (EI)	2	23.2	Reduction 1 g q8h (EI 3h)	Optimal	Yes	No
#4	35/M	Haematology + ICU	BSI	64	8 g LD 24 g/day CI	180.2	No	Cefiderocol 2 g q6h (EI)	8	0.9	Increase 2 g q6h CI	Quasi-optimal	Yes	No
#5	69/M	ICU	BSI + VAP	32	8 g LD 16 g/day CI	626.6	Reduction (12 g/day CI)	Cefiderocol 2 g q8h (EI)	2	6.3	No	Optimal	Yes	No
#6	79/M	ICU	VAP	32	8 g LD 16 g/day CI	458.3	No	CAZ-AVI 2.5 g q8h CI	8	14.9 (avibactam *f*C_ss_ 27.6 mg/L)	Reduction 1.25 g q8h CI	Optimal	Yes	Yes

AUC: area-under-the-curve; CAZ-AVI: ceftazidime-avibactam; CI: continuous infusion; C_min_: trough concentrations; C_ss_: steady-state concentration; EI: extended infusion; HAP: hospital-acquired pneumonia; ICU: intensive care unit; LD: loading dose; MIC: minimum inhibitory concentration; PK/PD: pharmacokinetic/pharmacodynamic; VAP: ventilator-associated pneumonia. Green box: achievement of optimal PK/PD targets (or microbiological eradication for microbiological outcome); yellow box: achievement of quasi-optimal PK/PD targets; red box: achievement of suboptimal PK/PD targets (or microbiological failure for microbiological outcome).

**Table 2 antibiotics-11-01739-t002:** Susceptibility profile of each DTR-PA clinical isolate and rationales for selecting combination therapy.

ID Cases (Combo)	Susceptibility Profile (MIC in mg/L)	Criteria for Combination Therapy
#1 CID + FOS	AMI ≤ 8; CEP > 8; CTZ 16; CTV > 8; CTT ≤ 1; CIP 1; IMI > 8; MER 32; PIT > 16; FOS 64; CID 1	(1) Ceftolozane-tazobactam supply shortage (2) Ceftazidime-avibactam resistance (3) Potential synergism of combination therapy (4) HAP
#2 CTV + FOS	AMI ≤ 8; CEP > 8; CTZ 16; CTV 8; CTT ≤ 1; CIP 1; IMI > 8; MER 32; PIT > 16; COL 2; FOS 256	(1) Ceftolozane-tazobactam supply shortage (2) Potential synergism of combination therapy according to in vitro evidence (3) VAP
#3 CID + FOS	AMI ≤ 8; CEP > 8; CTZ > 32; CTV > 8; CTT 4; CIP 0.5; IMI > 8; MER 32; PIT > 16; COL 2; FOS 32; CID 2	(1) Ceftolozane-tazobactam supply shortage (2) Ceftazidime-avibactam resistance (3) Potential synergism of combination therapy (4) VAP
#4 CID + FOS	AMI 16; CEP > 8; CTZ > 32; CTV > 8; CTT > 4; CIP > 1; IMI > 8; MER 16; PIT > 16; COL 1; FOS 64; CID 8	(1) Ceftolozane-tazobactam resistance (2) Ceftazidime-avibactam resistance (3) Potential synergism of combination therapy
#5 CID + FOS	AMI 16; CEP > 8; CTZ > 32; CTV > 8; CTT ≤ 1; CIP 1; IMI > 8; MER 32; PIT > 16; COL 1; FOS 32; CID 2	(1) Ceftolozane-tazobactam supply shortage (2) Ceftazidime-avibactam resistance (3) Potential synergism of combination therapy (4) VAP
#6 CTV + FOS	AMI > 16; CEP > 8; CTZ > 32; CTV 8; CTT > 4; CIP > 1; IMI 8; MER 8; PIT > 16; FOS 32	(1) Ceftolozane-tazobactam resistance (2) Potential synergism of combination therapy according to in vitro evidence (3) VAP

AMI: amikacin; CEP; cefepime; CID: cefiderocol; CTZ; ceftazidime; CTV: ceftazidime-avibactam; CTT: ceftolozane-tazobactam; CIP: ciprofloxacin; FOS: fosfomycin; HAP: hospital-acquired pneumonia; IMI: imipenem; MER: meropenem; PIT: piperacillin-tazobactam; VAP: ventilator-acquired pneumonia.

## Data Availability

The data that support the findings of this study are available on request to the corresponding author.

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
