# Peer review of "Pharmacokinetic/Pharmacodynamic Analysis of Continuous-Infusion Fosfomycin in Combination with Extended-Infusion Cefiderocol or Continuous-Infusion Ceftazidime-Avibactam in a Case Series of Difficult-to-Treat Resistant Pseudomonas aeruginosa Bloodstream Infections and/or Hospital-Acquired Pneumonia"

_antibiotics, 2022, doi:10.3390/antibiotics11121739_

Round 1

Reviewer 1 Report

Dear authors,

I have read the manuscript antibiotics-2043811 thoroughly. The submitted manuscript presents the results of a PK/PD study of continuous fosfomycin infusion and infusion of third-generation cephalosporins. The major limitation of the study is, as the authors also mention, very low number of participants. Overall, the quality of the study and the presentation of results are appropriate for a somewhat lower ranking magazine.

The quality of English language is very high and I am pleased, for providing the paper, which is easy to read and understand. The abstract is written intelligibly and clearly summarizes the content of the entire text. Keywords are appropriately chosen. In "Literature" part I noticed a large number of self-citations.

Introduction is concise and gives all the necessary information for readers. I would only draw attention to the last sentence of the first paragraph (Lines 54-55), which is written rather ambiguously.

 “Methods and materials” include all the necessary data to enable the reader to follow the described procedures. No improvement is needed in this part of a manuscript.

In the “Results and Discussion” part some of the numerical values are too precise regarding the number of participants.  I would suggest that the integer values are used instead of decimals.  For example, 5 out of 6 patients mathematically correspond to 83,3%, but the results of the study give a result that can be meaningfully interpreted in the range between 60 and 90 percent.

Regarding the decision to accept the contribution, I would suggest two options. The first is the inclusion of a larger number of subjects which will prolong the time until the publication of a high-quality paper. Another option is to publish a paper in a lower-ranked journal.

Kind regards.

Author Response

RESPONSE TO REVIEWERS

Manuscript ID: antibiotics-2043811 entitled “PK/PD analysis of CI fosfomycin in combination with EI cefiderocol or CI ceftazidime-avibactam in a case series of difficult-to-treat resistant Pseudomonas aeruginosa bloodstream infections and/or hospital-acquired pneumonia” by Gatti et al.

Dear Editor,

We would like to thank you for the opportunity to resubmit a revised version of this manuscript. We appreciated the reviewers’ constructive comments. All have been carefully considered and incorporated, where and whenever possible, in the revision.

Our point-by-point responses are provided below.

Q= QUERY; A= ANSWER

Reviewer #1

Q1. I have read the manuscript antibiotics-2043811 thoroughly. The submitted manuscript presents the results of a PK/PD study of continuous fosfomycin infusion and infusion of third-generation cephalosporins. The major limitation of the study is, as the authors also mention, very low number of participants. Overall, the quality of the study and the presentation of results are appropriate for a somewhat lower ranking magazine.

The quality of English language is very high and I am pleased, for providing the paper, which is easy to read and understand. The abstract is written intelligibly and clearly summarizes the content of the entire text. Keywords are appropriately chosen. In "Literature" part I noticed a large number of self-citations.

A1. We thank the reviewer for appreciating our work. We agree that the sample size is quite low and this is a major limitation of the study, as we recognized in the Discussion section. However, it should also be considered that this is the first clinical case series in which the relationship between the PK/PD profiles of CI fosfomycin combined with EI cefiderocol or with CI ceftazidime-avibactam and the microbiological outcome of DTR-PA infections was assessed. We agree also that there are some self-citations, even if overall they accounted for more or less 10% of the references. Besides this, the rationales for this choice were based on the fact that our centre is deeply engaged in adopting some innovative policies of antimicrobial stewardship, which are recognized in the literature. We used them for citing the advantages of continuous infusion compared to intermittent infusion (reference no. 16), the role of TDM-guided strategy for optimizing the use of novel beta-lactams (references no. 20-21), the potential advantage of combining fosfomycin with novel BLs (reference no. 32), the need of maximizing the PK/PD targets of antibiotics for minimizing resistance occurrence (reference no. 46) and for adopting proper strategies of dosing adaptation (reference no. 48).

Q2. Introduction is concise and gives all the necessary information for readers. I would only draw attention to the last sentence of the first paragraph (Lines 54-55), which is written rather ambiguously.

A2. Thank you for this comment. As suggested, we reformulated the sentence to improve readability.

Q3. “Methods and materials” include all the necessary data to enable the reader to follow the described procedures. No improvement is needed in this part of a manuscript.

A3. Thank you for your appreciation.

Q4. In the “Results and Discussion” part some of the numerical values are too precise regarding the number of participants.  I would suggest that the integer values are used instead of decimals.  For example, 5 out of 6 patients mathematically correspond to 83,3%, but the results of the study give a result that can be meaningfully interpreted in the range between 60 and 90 percent.

A4. Thank you for your suggestion. We replaced decimal values with integer numbers in Results section.

Q5. Regarding the decision to accept the contribution, I would suggest two options. The first is the inclusion of a larger number of subjects which will prolong the time until the publication of a high-quality paper. Another option is to publish a paper in a lower-ranked journal.

A5. As reported in comment no. 1, we agree that the sample size is quite low and this is a major limitation of the study, as we recognized in the Discussion section. However, it should also be considered that this is the first clinical case series in which the relationship between the PK/PD profiles of CI fosfomycin combined with EI cefiderocol or with CI ceftazidime-avibactam and the microbiological outcome of DTR-PA infections was assessed. As far as the possibility of increasing the sample size is concerned, we could argue that the time needed for reaching this could be very long and unpredictable. Nowadays, ceftolozane-tazobactam is the first-line treatment of DTR-PA infections, and fortunately the no. of DTR-PA strains is quite limited. Indeed, in our series only 2/6 cases were related to ceftolozane-tazobactam resistance. Conversely, in 4/6 cases the strategy of adopting CI fosfomycin + EI cefiderocol or CI ceftazidime/avibactam was the consequence of the unpredictable long-term supply shortage of ceftolozane-tazobactam that occurred during last year and currently is completely resolved. For these reasons, we believe that the possibility of increasing significantly the sample size in a reasonable timeframe it could be quite limited.

Reviewer 2 Report

The authors present a case series of six patients with difficult to treat Pseudomonas aeruginosa infection. These patients were treated with fosfomycin continuous infusion + ceftazidime-avibactam continuous infusion or plus cefiderocol extended infusion. Therapeutic drug monitoring was performed and the outcome in relation to the pk/pd-target attainment is reported.

The manuscript is well written. Since real world data on pk/pd targets in this difficult to treat patient population is scarce the case series is interesting. The discussion is overall adequate.

There is one point that could be added to the discussion and/or data presentation: The aim of TDM is to ensure effective and safe treatment, which is also stressed by the authors. In table 1 it is evident or suggested that TDM led to dosage adjustment (3, 4, 5 and 6). Dosage increase of Cefiderocol in patient 4 is evident, because the pk/pd target was not attained. In the other cases a dose reduction was performed. It would be interesting if the authors could discuss the reason  - is there evidence of a pk/pd threshold for toxic effects? Or were there signs of concentration dependent adverse effects in the patients? Why was no increase of fosfomycin dosage performed in patient 2?

Author Response

RESPONSE TO REVIEWERS

Manuscript ID: antibiotics-2043811 entitled “PK/PD analysis of CI fosfomycin in combination with EI cefiderocol or CI ceftazidime-avibactam in a case series of difficult-to-treat resistant Pseudomonas aeruginosa bloodstream infections and/or hospital-acquired pneumonia” by Gatti et al.

Dear Editor,

We would like to thank you for the opportunity to resubmit a revised version of this manuscript. We appreciated the reviewers’ constructive comments. All have been carefully considered and incorporated, where and whenever possible, in the revision.

Our point-by-point responses are provided below.

Q= QUERY; A= ANSWER

Reviewer #2

The authors present a case series of six patients with difficult to treat Pseudomonas aeruginosa infection. These patients were treated with fosfomycin continuous infusion + ceftazidime-avibactam continuous infusion or plus cefiderocol extended infusion. Therapeutic drug monitoring was performed and the outcome in relation to the pk/pd-target attainment is reported.

The manuscript is well written. Since real world data on pk/pd targets in this difficult to treat patient population is scarce the case series is interesting. The discussion is overall adequate.

We thank the reviewer for the appreciation.

Q1. There is one point that could be added to the discussion and/or data presentation: The aim of TDM is to ensure effective and safe treatment, which is also stressed by the authors. In table 1 it is evident or suggested that TDM led to dosage adjustment (3, 4, 5 and 6). Dosage increase of Cefiderocol in patient 4 is evident, because the pk/pd target was not attained. In the other cases a dose reduction was performed. It would be interesting if the authors could discuss the reason  - is there evidence of a pk/pd threshold for toxic effects? Or were there signs of concentration dependent adverse effects in the patients? Why was no increase of fosfomycin dosage performed in patient 2?

A1. We thank the reviewer for this comment, allowing us to better clarify this important issue. As reported in the Results section (refer to Line 135), no treatment-related adverse events emerged. Although a specific PK/PD threshold for toxicity was not yet defined for novel beta-lactams, the risk of neurotoxicity cannot be ruled out according to the relationship existing with high exposures reported for traditional cephalosporins (refer to Vardakas et al., doi: 10.1080/14740338.2018.1462334). We clarified this point in Discussion section (refer to Line 162-166). As regard patient no. 2, the reason for not increasing fosfomycin dosing was related to the severe worsening of clinical conditions due to the concomitant invasive pulmonary aspergillosis and the consequent need of withdrawal of life-sustaining treatment.

Round 2

Reviewer 1 Report

Dear Authors,

Last time I assessed that, in my opinion, the quality of the contribution was not acceptable for such a highly ranked journal as Antibiotics from MDPI. In my opinion, the study should be significantly broader and include more subjects from different locations. In its current form, in my opinion, it would be more suitable for publication in a journal with a JCR IF of around 2-3. However, the article is written clearly and comprehensibly, the changes made by the authors are welcome, there are no methodological ,mistakes, so I will not particularly object to the decision of the other reviewers and the journal's editorial board if they decide the article is appropriate for the special issue of "Antibiotics"..

Best regards.